# Hierarchical triphase diffusion photoelectrodes for photoelectrochemical gas/liquid flow conversion

Xiangyu Meng[1,6], Chuntong Zhu[1,6], Xin Wang[2,6], Zehua Liu[3], Mengmeng Zhu[1], Kuibo Yin[4], Ran Long [3], Liuning Gu[5], Xinxing Shao[5], Litao Sun [4], Yueming Sun[1], Yunqian Dai [1] ✉ & Yujie Xiong [2,3] ✉

Photoelectrochemical device is a versatile platform for achieving various chemical transformations with solar energy. However, a grand challenge, originating from mass and electron transfer of triphase—reagents/products in gas phase, water/electrolyte/products in liquid phase and catalyst/photoelectrode in solid phase, largely limits its practical application. Here, we report the simulation-guided development of hierarchical triphase diffusion photoelectrodes, to improve mass transfer and ensure electron transfer for photoelectrochemical gas/liquid flow conversion. Semiconductor nanocrystals are controllably integrated within electrospun nanofiber-derived mat, overcoming inherent brittleness of semiconductors. The mechanically strong skeleton of free-standing mat, together with satisfactory photon absorption, electrical conductivity and hierarchical pores, enables the design of triphase diffusion photoelectrodes. Such a design allows photoelectrochemical gas/liquid conversion to be performed continuously in a flow cell. As a proof of concept, 16.6- and 4.0-fold enhancements are achieved for the production rate and product selectivity of methane conversion, respectively, with remarkable durability.

Photoelectrochemical (PEC) devices, which perfectly combine the advantages of photocatalysis and electrocatalysis, provide a class of promising platforms for driving a variety of energy-related chemical transformations including but not limited to solar-driven water splitting, carbon dioxide reduction, nitrogen fixation and methane conversion. Universally, gas–liquid–solid triphase is involved in almost all the PEC chemical transformations. The central components of PEC devices are solid catalysts and photoelectrodes immersed in liquid electrolyte. To sustain the aforementioned chemical reactions, reagents (i.e., $H_2O$, $CO_2$, $N_2$, $CH_4$) in gas or liquid phase should be

efficiently transported to the solid surface, while products (i.e., $H_2/O_2$, gas or liquid carboneous products, $NH_3/NH_4^+$) have to be diffused out and released from the solid phase to gas or liquid phase. Particularly, the low solubility of related gaseous molecules in water results in a huge obstacle for mass transfer. As such, the triphase mass transfer has been a long-standing bottleneck for scaling up PEC chemical transformations with required high reaction rate and flux[1,2]. Most recently, continuous flow cells have been developed for electrochemical $CO_2$ reduction, water splitting and other applications by designing gas diffusion electrodes[3–5], which can overcome the mass

[1]School of Chemistry and Chemical Engineering, Southeast University, Nanjing, Jiangsu 211189, China. [2]Anhui Engineering Research Center of Carbon Neutrality, School of Chemistry and Materials Science, Anhui Normal University, Wuhu, Anhui 241000, China. [3]School of Chemistry and Materials Science, Hefei National Laboratory for Physical Sciences at the Microscale, and National Synchrotron Radiation Laboratory, University of Science and Technology of China, Hefei, Anhui 230026, China. [4]School of Electronic Science and Engineering, Southeast University, Nanjing, Jiangsu 211189, China. [5]School of Civil Engineering, Southeast University, Nanjing, Jiangsu 211189, China. [6]These authors contributed equally: Xiangyu Meng, Chuntong Zhu, Xin Wang. ✉e-mail: daiy@seu.edu.cn; yjxiong@ustc.edu.cn

transfer limitation that batch cells used to suffer from. However, the flow cell configuration for PEC chemical transformations is substantially more complicated, setting up an all-in-one requirement for photoelectrodes—high photon absorption, electrical conductivity, gas permeability and catalytic activity on semiconductor materials. This high requirement poses a grand challenge to the design of triphase diffusion photoelectrodes, because usually high-performance semiconductors are in the form of dense powders or layers that can hardly be supported on gas-permeable substrates.

Intuitively, free-standing fibrous mat made of suitable semiconductor, which offers sufficient mass transfer channels and fully exposed surfaces, should be an excellent candidate for tackling this challenge. If such a material can be fabricated, the semiconductor will simultaneously serve as light absorber, PEC catalyst and mass diffusion layer, without use of any additives or substrates. Certainly, this class of diffusion photoelectrodes can also effectively enhance system durability by circumventing the falloff problem of conventional catalyst loading, as well as improve efficiency by rescuing photons or electrons from additives/substrates loss. Electrospinning is a widely used approach to free-standing fibrous mats. However, when the rapid fluid and gas flows are implemented to the flow cell for achieving PEC gas/liquid conversion with high reaction rates, the structures of triphasic interfaces and pores/channels should be repellent to high pressure. Recently, progress has been made in improving the mechanical strength and deformability of electrospun oxide nanofiber mat, by inducing pre-strain in curved nanofibers, crosslinking sol-gels or assembling nanofibers into topological structures[6–8]. It still remains challenging to enhance both mechanical strength and flexibility based on inherently fragile semiconductor oxide crystal structures without adding carbon-containing polymers as binders[9,10]. More importantly, semiconductor oxide nanofibers should ideally possess a straight and directional structure, which can shorten diffusion paths for rapid mass and electron transfer in PEC flow reactions. Due to the compensation of inherent deformation along or among straight nanofibers, this demands higher requirements for achieving high mechanical stability of fibrous mats.

Here, we report the triphase diffusion photoelectrodes in a PEC flow cell of continuous gas/liquid conversion. To better demonstrate our concept, we selected methane conversion as a model system considering the particularly low solubility of $CH_4$ in water that significantly limits the mass transfer to catalyst/photoelectrode surface. Methane, which is abundant in natural gas, shale gas and combustible ice, is a viable alternative to crude oil in terms of producing energy or vital chemicals[11–13]. The long-distance transportation and liquefaction of low-boiling $CH_4$ gas from remote extraction areas (e.g., epeiric sea) are energy-intensive, with a major concern of $CH_4$ leakage given the 25-fold greater greenhouse effect than $CO_2$[14]. Hence, the PEC conversion of $CH_4$ into value-added products, which may be conducted exactly at $CH_4$ extraction location instead of conventional $CH_4$ combustion, transportation or storage, is technologically worthy of development. Given that PEC $CH_4$ conversion should be performed on photoanode, we specifically designed free-standing $TiO_2/ZnWO_4$ (i.e., *n*-type semiconductor) fibrous mats. Guided by material simulations, the triphase diffusion mat was designed with hierarchical fibrous pores for facilitating mass transfer and with amphoteric interfaces for circumventing $CH_4$ dissolution limitation while ensuring adequate aqueous electrolyte diffusion. To accomplish the design, mechanically strong $TiO_2/ZnWO_4$ nanofibers were fabricated by refining nanocrystals and inducing high-density dislocations (~$10^{10}$ mm$^{-2}$) inside each nanofiber, which then can be facilely interwoven into a flexible mat without any cracking even under high-speed (100 mL/min) fluid flow. The inimitable merits of our designed triphase diffusion photoanodes enabled working in a PEC flow cell for continuous $CH_4$ conversion, which can largely overcome the bottlenecks of traditional PEC batch

cells, to achieve the enhancement of production rate and $C_2$ product selectivity by 16.6 times and 4.0 times, respectively. Remarkably, the designed triphase diffusion photoanodes offered excellent stability for at least 100 h. This work provides a strategy of free-standing structural design to promote mass transfer at triphasic interfaces, which should be universal for various chemical transformations.

## Results

### Design of hierarchical $TiO_2/ZnWO_4$ fibrous mat

To enhance triphase diffusion, we employed molecular dynamics (MD) and calculation fluid dynamics (CFD) simulations to design interwoven nanofibers with a focus on the triphasic interface and pores/channels of mass transfer. The ultimate goal of this design is to achieve the maximum contact of $CH_4$, water/electrolyte and catalyst/photoanode through rapid multiphasic fluid flow. Figure 1a illustrates the multi-level pore structures formed between interwoven nanofibers at microscale and inside single nanofiber at nanoscale, respectively. Interestingly, these pores can remarkably enhance mass transfer and even change flow behavior. According to Bernoulli's principle in fluid dynamics[15], flow velocity dramatically increases with constriction of flow volumes. Our CFD simulation results (Fig. 1b) revealed the change of flow velocity from -0.045 m/s to -0.285 m/s as the fluid flowed from reactor chamber to interwoven pores among nanofibers. The detailed models and liquid fluid fraction distributions are shown in Supplementary Fig. 1 and 2, respectively. Darcy's law indicates that such a localized promotion effect for mass transfer velocity in the porous media within interwoven mat can be further amplified with the increase of flow rate in flow cell[16].

The simulations further revealed the change of turbulence flow behavior with clear vortex flow lines when gas and liquid were mixed and diffused within the interwoven pores of fibrous mat (Fig. 1c). In contrast, the flow lines in the reactor chamber underwent nearly directly from inlet to outlet (i.e., mat), showing a typical laminar flow behavior with smaller velocities. This behavior change from laminar flow to turbulent flow, occurring at fibrous pores, was induced by the forced convection in fibrous pores and the enhanced shear force at $CH_4/H_2O$ interface when fluids collided among pore walls. Hence, the interwoven pore structures within the fibrous mat are beneficial to achieving continuous, rapid and sufficient flow diffusion of $CH_4$ and electrolyte on photoanode.

To improve the accessibility of $CH_4$ to catalyst in aqueous electrolyte, simulations suggested that the ampholytic interfaces with affinities for both liquid water and gas $CH_4$ should be constructed on the nanopores inside each nanofiber. The right scheme in Fig. 1a illustrates the interface design concept. Specifically, the nanopore interface should have a low surface energy (i.e., hydrophobic) side to promote the continuous accessibility of undissolved $CH_4$ to nanopore, by minimizing resistance of $CH_4$ transfer in aqueous electrolyte. Meanwhile, a high surface energy (i.e., hydrophilic) side should be built to ensure the availability of protons from aqueous electrolyte, increasing electron conductivity. Furthermore, the high surface energy side on the nanopore can prevent the formation of large $CH_4$ gas bubbles, potentially reducing the shielding effect on interface during a long-term operation. The gas/liquid distribution on the interface of a nanopore inside each nanofiber was predicted by static MD simulation (Fig. 1d). The densities of $CH_4$ and water, derived from dynamic statistical MD simulation, were also demonstrated in Fig. 1d. Enabled by this structural design, the contact of $CH_4$ and electrolyte to the porous walls in catalyst can be realized with a concentrating effect (highlighted by black arrows in Fig. 1d). In this context, we can employ the electrospun $TiO_2/ZnWO_4$ fibrous mat with these predesigned interface and pore features to construct triphase diffusion photoanodes, which will allow investigating multiphasic flow chemistry and achieving efficient PEC $CH_4$ conversion to value-added products in flow cells.

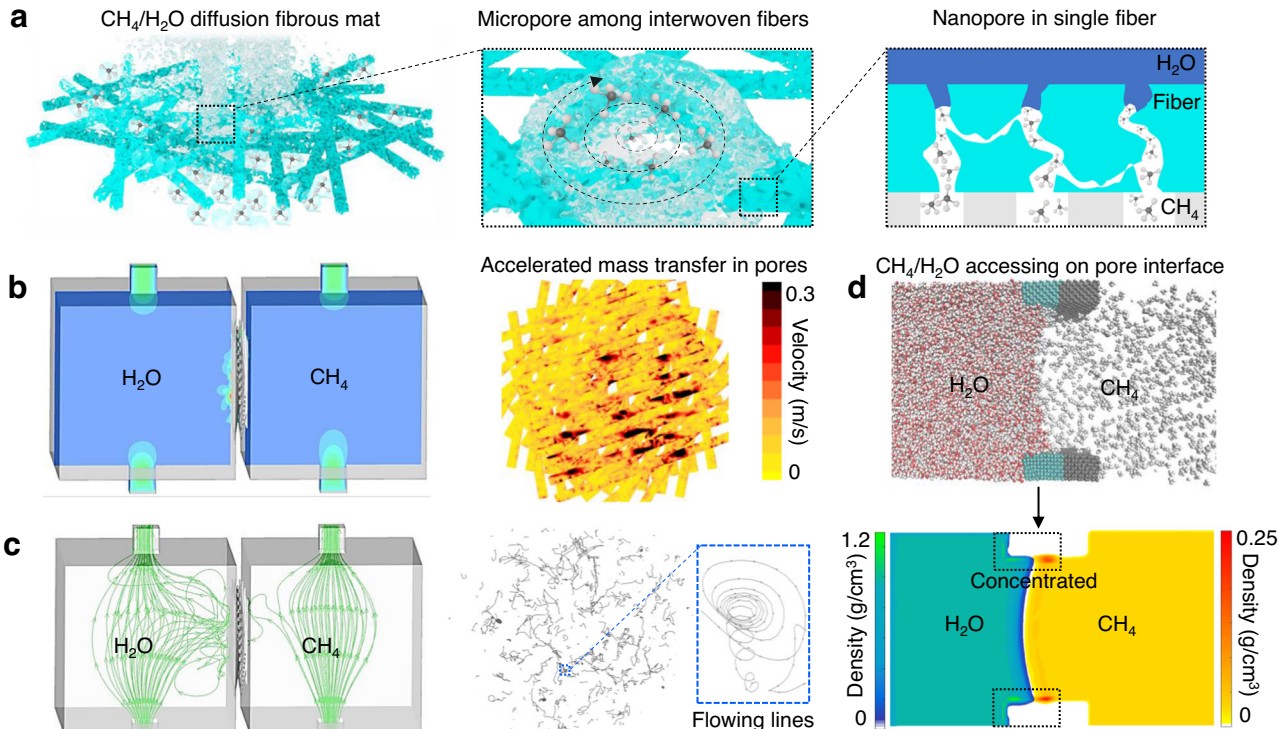

**Fig. 1 | Structural design of hierarchical fibrous mat in PEC flow cell for CH$_4$ conversion. a** Schematic illustration of fibrous mat with hierarchical pore structures for CH$_4$/H$_2$O diffusion. **b** CFD simulation showing the diffusion velocities of CH$_4$ gas and electrolyte liquid in reactor (left) and on micropores among interwoven nanofibers (right). The color bar in (**b**) represents the velocity (m/s). **c** CFD simulation showing the flow streamlines of CH$_4$ gas and electrolyte liquid in reactor (left) and on micropores among interwoven nanofibers (right). The blue dashed boxes highlight the areas of amplification. **d** MD simulation results showing the distribution (top) and density (bottom) of CH$_4$ gas/H$_2$O liquid at interface of nanopores inside a single nanofiber. The black dashed boxes highlight the locally concentrated CH$_4$ and H$_2$O on the designed pore interfaces. The color bars in (**d**) represent the densities of molecules (g/cm$^3$).

## Fabrication of hierarchical TiO$_2$/ZnWO$_4$ fibrous mat

Guided by the simulation design, we developed an electrospinning −calcination approach to fabrication of TiO$_2$/ZnWO$_4$ nanofibers with targeted structures. Typically, each nanofiber showed highly porous structures with uniformly dispersed elements, as shown in scanning transmission electron microscopy (STEM) images and energy-dispersive X-ray spectroscopy (EDS) elemental mapping profiles (Fig. 2a). In addition to the observation of pores from space projection, the advanced focused ion beam (FIB) together with transmission electron microscopy (TEM) were used to assist the examination of pores inside a nanofiber. After three-dimensional (3D) reconstruction based on FIB-TEM slice information, interconnected pore channels with an average porosity of 34% were clearly observed within a single nanofiber (Fig. 2b). These interconnected nanopores inside each nanofiber lay the foundation for enabling quick mass transfer and forming gas-accessible interfaces.

The detailed structures of nanofibers are further examined by high-resolution TEM (HRTEM) and geometric phase analysis (GPA) (Fig. 2c−f). As verified by HRTEM (Fig. 2c, e) and X-ray diffraction (XRD) (Supplementary Fig. 3) analysis, the thin nanofibers with average diameter of 109 nm are composed of well-defined anatase TiO$_2$ and monoclinic ZnWO$_4$ nanocrystals. HRTEM images show the typical TiO$_2$ {101} facets and ZnWO$_4$ {010} facets with clear crystal boundaries (highlighted by orange lines) between the two phases (Fig. 2c) or within a single phase (Fig. 2e)[17,18]. Surprisingly, a high density of dislocations (~10$^{10}$ mm$^{-2}$, the average number of dislocations per unit area) near these crystal boundaries can be found (highlighted by white box in HRTEM images)[19,20]. The strain distribution at the boundaries was then calculated by GPA. The strain mapping (Fig. 2d, e) revealed that the compression−tension strain pairs exactly occurred at the dislocation areas. This finding indicated that the dislocations were

formed by the strains at crystal boundaries. The one-dimensional confinement at nanoscale, observed in the structure of the ultrathin nanofibers, can restrict oxide crystal overgrowth refining the crystal size. Under the radial-confined growth, the oxide crystals were arranged more compactly enhancing displacement force on their boundaries in the nonequilibrium condition at high temperature[21,22]. This growth mode, which is distinct from other crystal assembly methods[19,20,23], can lead to high-density inherent dislocations inside each nanofiber. The dislocations can reduce the oxides' brittleness[23]. The GPA results demonstrated that the strain field surrounding dislocations can enhance the fracture toughness via the stress shielding effect, which typically necessitates a dislocation density of at least 10$^8$ mm$^{-2}$ in bulk oxides[23]. Our high-density dislocations in TiO$_2$/ZnWO$_4$ oxide achieved by electrospinning method make such beneficial features come true.

In addition to mechanical strength, optical absorption and electrical conductivity are other prerequisites for our photoanode application. Semiconductor oxide commonly has limited ability of optical absorption in a broad spectrum as well as unfavorable electrical conductivity[24]. To meet the optical and electrical demands in PEC CH$_4$ conversion, the TiO$_2$/ZnWO$_4$ nanofibers were reduced to generate oxygen vacancies (OVs) (Supplementary Fig. 4). Electron paramagnetic resonance (EPR) spectra were collected in dark to examine the generated OVs in terms of spin electrons (Supplementary Fig. 5). The solitary Lorentz peak at $g = 2.003$ in the spectrum of TiO$_2$/ZnWO$_4$ was observed and attributed to the electrons in conduction band (CB). For reduced-TiO$_2$/ZnWO$_4$ (namely, R-TiO$_2$/ZnWO$_4$), the signal of this peak was markedly amplified, indicating a higher electron density in CB and unbound electrons trapped by OVs[25]. X-ray photoelectron spectroscopy (XPS) confirmed the purity of R-TiO$_2$/ZnWO$_4$ after the reduction (Supplementary Fig. 6). High-resolution O1$s$ XPS spectra of

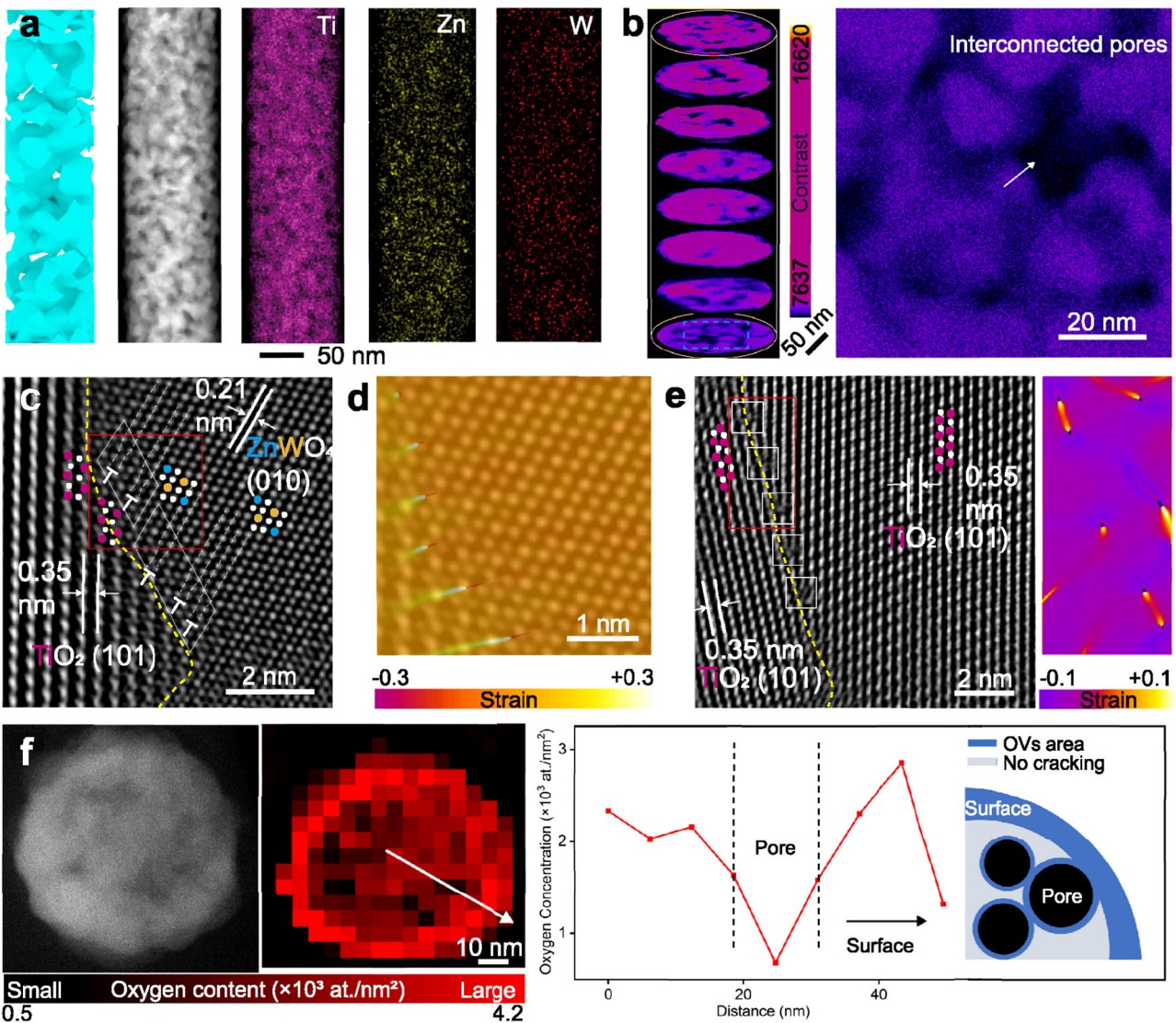

**Fig. 2 | Structural characterization of porous nanofibers. a** Structural illustration, STEM image and corresponding EDS elemental mappings of a nanofiber. **b** Three-dimensional reconstruction and TEM image of FIB slice showing the interconnected pores (highlighted by the arrow) inside a nanofiber. The area of amplification in (**b**) is marked with a blue dashed box. The color bar in (**b**) represents the contrast. HRTEM images (**c, e**) and corresponding GPA strain analysis (**d, e**) of grain boundaries within a nanofiber. In (**c**) and (**e**), strain analysis area is marked with red boxes; Ti, O, Zn and W atoms are highlighted by magenta red, white, blue and orange dots, respectively; dislocation areas are marked with white boxes; grain boundaries are highlighted by yellow dashed lines; lattice spacing is highlighted by white lines in pairs. The color bars in (**d**) and (**e**) represent the strain. **f** STEM image and corresponding EELS mapping of oxygen atoms on the FIB slice of nanofiber (left), and the analysis results of oxygen content distribution on a nanofiber slice (right). The color bar in (**f**) represents the oxygen content (×10³ at./nm²) on slice.

R-TiO$_2$/ZnWO$_4$ revealed the lattice oxygen, OVs and adsorbed moisture binding peaks at 529.72, 531.55 and 533.65 eV, respectively. It further indicated the existence of OVs and was consistent with the finding of the EPR test[26]. In contrast, TiO$_2$/ZnWO$_4$ before reduction showed a content of 30.34% at 531.55 eV pointed to OVs, substantially lower than that of R-TiO$_2$/ZnWO$_4$ (58.32%).

It is worth pointing out that mechanical strength is commonly compromised when enhancing optical and electrical properties through reduction (e.g., high-temperature reduction, wet-chemical reduction)[25–27]. In our case, benefiting from interconnected pore channels, the reductant was mostly transferred to and reacted with the nanofiber surface and the pore walls, thus remaining a rather intact and strong fibrous skeleton. Electron energy loss spectroscopy (EELS) mapping (Fig. 2f), which can resolve the oxygen distribution within a typical slice of porous nanofiber, revealed that the effective oxygen reduction took place locally along pore surfaces while remaining intact oxide skeleton. Collecting data from different slices of R-TiO$_2$/ZnWO$_4$ nanofiber (Supplementary Fig. 7), EELS mapping showed the same trends, further confirming the reduction locations. The reduction process, locally on pore surfaces where mass transfer occurred, is beneficial for acquiring both activity and mechanical stability.

Upon acquiring structural information for nanofibers, the mechanical properties especially under the real gas/liquid flow conditions, which are vital toward realizing the predesigned interface and mass transfer behavior in flow cells, were closely examined. In situ TEM images recorded the mechanical bending of a single nanofiber under enlarged deformation angles without any observed cracking (Fig. 3a). In the meantime, the fracture surface of nanofiber after cracking was observed by TEM (Fig. 3b and Supplementary Fig. 8), showing residual dislocation-growth behavior along the strain directions. This result further confirmed that the dislocation-growth distribution suppressed the direct crack growth under external strain. Further, the

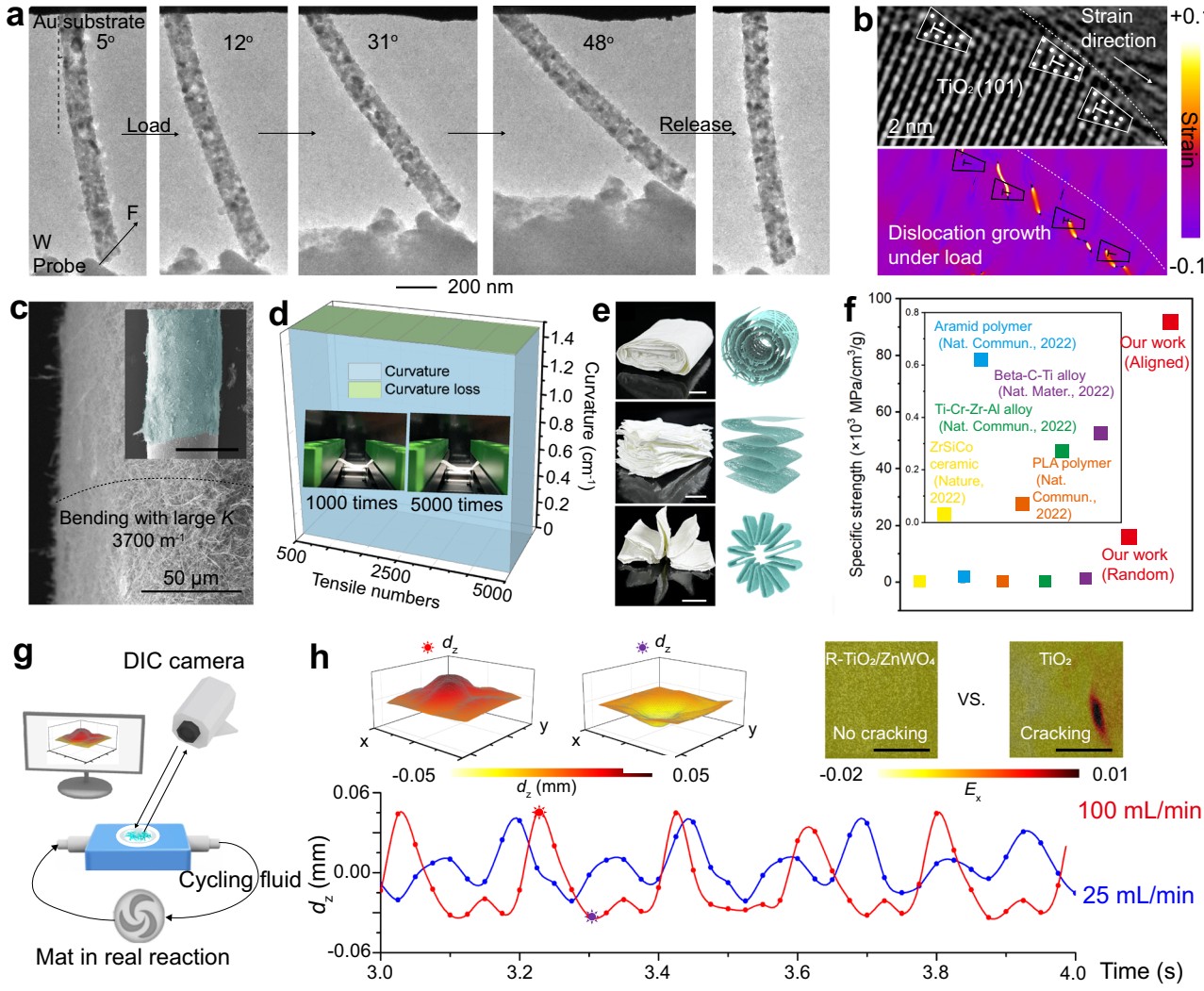

**Fig. 3 | Mechanical characterization of fibrous mat. a** In situ TEM observation of a single nanofiber under bending deformations. **b** TEM image showing the crack section of crystal under strain direction, with GPA strain analysis (inset). The color bar in (**b**) represents the strain. **c** SEM image of fibrous mat wrapped on an ultrathin tube with different magnifications. The scale bar in the inset SEM image is 1 mm. **d** Stability of mat over 5000-cycle bending with angle over 60°. **e** Optical images and corresponding schemes of the flexible $TiO_2/ZnWO_4$ mat with various origami structures. The scale bars represent 1 cm. **f** Comparison of specific mechanical strength among our work and recently advanced works. **g** Schematic illustration of DIC test for measuring micro-strain within a R-$TiO_2/ZnWO_4$ mat in the presence of flowing $CH_4$ gas or electrolyte liquid. **h** DIC images (top) and real-time micro-strain curves (bottom) of the R-$TiO_2/ZnWO_4$ mat by cycling $CH_4$/electrolyte with a flow rate of 100 mL/min. The DIC images show the absence of cracking in our mat while the conventional $TiO_2$ mat has cracks. The scale bars in (**h**) are 0.5 cm. The color bars in (**h**) represent the $d_z$ (mm) detected in DIC tests. The red and purple symbol in (**h**) marked the peak and valley in curve, respectively.

mechanically strong nanofibers were assembled into interwoven mat by electrospinning without binders. The resultant free-standing and flexible mat is still mechanically strong so that it can be easily wrapped on a thin tube ($d = 1$ mm). Scanning electron microscopy (SEM) image shows that the intact structure with an ultra-large bending curvature was achieved by our oxide fibrous mat (Fig. 3c). This mechanically strong and flexible mat can maintain stability even after 5,000 times bending over 60° (Fig. 3d and Supplementary Video 1). Moreover, the mat can be easily deformed into various 3D electrode structures by easy origami folding (Fig. 3e). Notably, the fibrous mats had an ultra-high specific strength over 90,000 MPa/cm³/g as calculated from stress-strain curves (Fig. 3f and Supplementary Fig. 9), outperforming many recently reported advanced works[28–31]. R-$TiO_2/ZnWO_4$ showed a tensile strength of 1.19 MPa and a Young's modulus of 62.2 MPa, as well as a tensile strain of 1.84% (Supplementary Fig. 10a), indicating that reduction process had little influence on the mechanical properties of $TiO_2/ZnWO_4$. As the real pressure during fluid flowing can also be in compression mode, R-$TiO_2/ZnWO_4$ compression tests were carried out (Supplementary Fig. 10b). The sample showed a compressive strength of 0.0761 MPa and a Young's modulus of 3.52 MPa, with a satisfying ductility strain of 9.34%. The unit rupture work of R-$TiO_2/ZnWO_4$ in compression was as high as 3100 J m⁻³ g⁻¹, outperforming previously reported advanced ceramic materials[32–34].

To further assess the mechanical stability, advanced digital image correlation (DIC) technology was conducted under practical gas/liquid fluid flow conditions in a flow cell, as schematically illustrated in Fig. 3g (details of the device can be found in Supplementary Fig. 11 and "Methods" section). During the high fluid flow rate up to 100 mL/min, the real-time micro-strain applied on R-$TiO_2/ZnWO_4$ fibrous mat was in situ recorded (Fig. 3h), showing periodic deformation caused by fluid pumping. We further analyzed micro-strain under fluids with different flow rates and operation time (Supplementary Fig. 12). Two typical "peak" and "valley" of micro-strain distributed on fibrous mat surface with a fluid flow rate of 100 mL/min were also demonstrated by 3D displacement reconstruction (Fig. 3h). During the cycling fluid flow, our mat remained intact without cracking all the time. In contrast, the

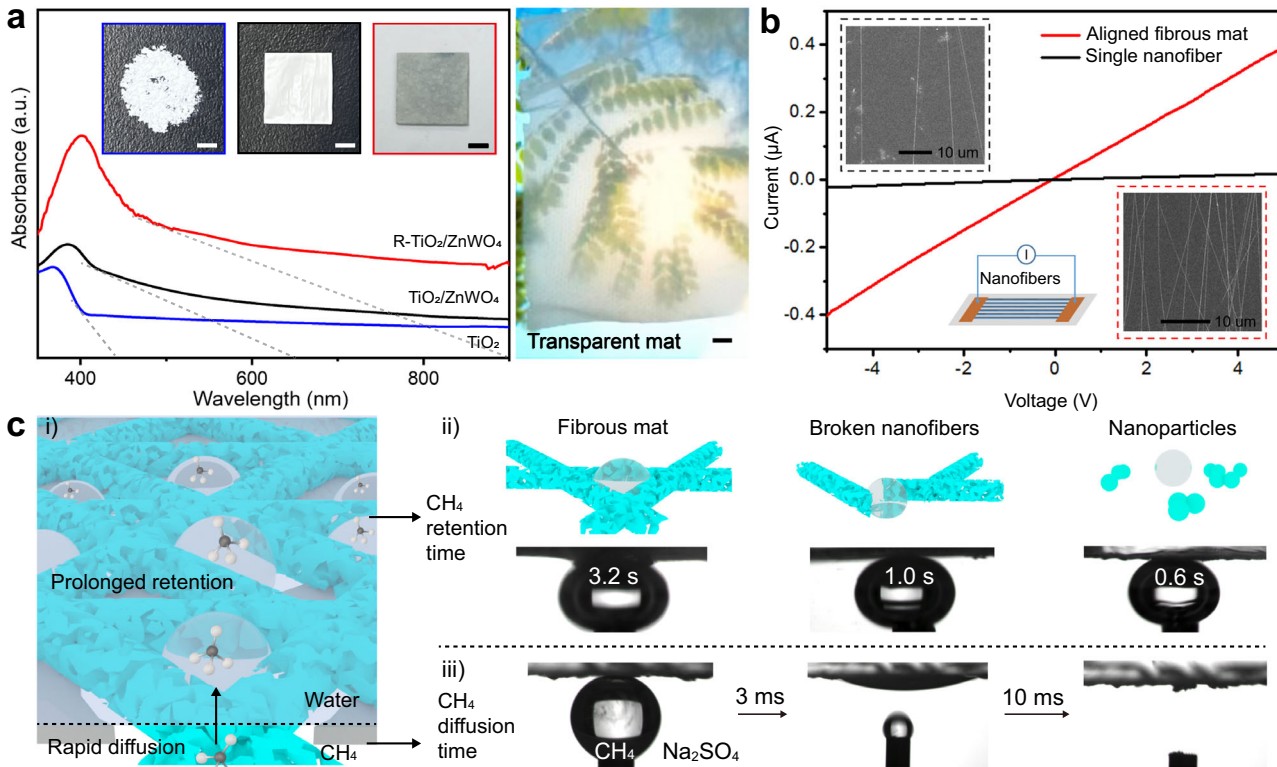

**Fig. 4 | Optical, electrical and triphasic interfacial properties of fibrous mat.**
**a** UV–vis absorption spectra of R-TiO$_2$/ZnWO$_4$, TiO$_2$/ZnWO$_4$ and TiO$_2$. Insets in spectra show the optical images of non-free-standing TiO$_2$, free-standing TiO$_2$/ZnWO$_4$ and R-TiO$_2$/ZnWO$_4$ mat. The optical image (right) shows a transparent TiO$_2$/ZnWO$_4$ fibrous mat. The scale bars represent 5 mm in (**a**). **b** The inherent conductivity test of R-TiO$_2$/ZnWO$_4$ with and without fibrous interwoven structure. The inset SEM images and schemes illustrate the fibrous structures and the test method, respectively. **c** Schematic illustration of the triphasic interfaces in the designed R-TiO$_2$/ZnWO$_4$/PTFE mat (i). The contact angles of CH$_4$ gas bubbles on the surfaces of R-TiO$_2$/ZnWO$_4$ fibrous mat (ii, left), on broken nanofibers (ii, middle), and on nanoparticles (ii, right) in the same Na$_2$SO$_4$ (pH = 2) liquid. The inserted schemes illustrate these structures. The contact angle of CH$_4$ gas bubbles on the surface of PTFE shows the rapid gas diffusion into R-TiO$_2$/ZnWO$_4$ (iii).

conventional TiO$_2$ fibrous mat was cracked after rapid fluid flow through the mat. This striking contrast indicated that the strength and toughness of our fibrous mat has improved significantly. As such, it can meet the significant demand of diffusion photoanode in continuous flow reactions.

Given that the mechanical requirement for flow CH$_4$ conversion can be met by our design, this free-standing fibrous mat was further evaluated with light absorption and electrical conductivity. UV–vis absorption spectra were collected to evaluate the light absorption of representative mats (free-standing R-TiO$_2$/ZnWO$_4$, TiO$_2$/ZnWO$_4$, and non-free-standing TiO$_2$) (Fig. 4a). All three samples exhibited the similar light absorption band with an onset at *ca*. 400 nm. The addition of ZnWO$_4$ nanocrystals to nanofibers broadened the range of light absorption. Meanwhile, the defective structure with OVs in R-TiO$_2$/ZnWO$_4$ resulted in a further red-shift of the tail adsorption up to 900 nm, covering the entire visible-light range. The direct bandgap of TiO$_2$, TiO$_2$/ZnWO$_4$ and R-TiO$_2$/ZnWO$_4$ samples calculated by Tauc plot method from the light absorption spectra was 3.18, 2.51 and 2.40 eV, respectively[35]. In addition to the broadening of light absorption by OVs in nanocrystal lattice, the inherent fibrous assembly structure within mat enabled a transparency ratio of 93% (calculated from UV–vis absorption spectra in Supplementary Fig. 13, see structures in Supplementary Fig. 14). This feature will allow high-flux light to travel through the mat, as shown in the right panel of Fig. 4a. Based on the transparency of free-standing oxide mat, it can be used as both gas diffusion layer and sun light absorber, free of troubling problems originally in densely layered catalysts. This feature also offers the opportunity of customizing photoanode structures without concerns on light incidence and gas diffusion directions in flow cells.

To assess electrical conductivity, R-TiO$_2$/ZnWO$_4$ nanofiber was then examined through current–voltage (*I–V*) curves (details in Supplementary Fig. 15). The conductivity of R-TiO$_2$/ZnWO$_4$ nanofiber calculated from *I–V* curve (Fig. 4b) was 0.0215 S/cm, over 25 times higher than that of TiO$_2$/ZnWO$_4$ nanofiber (0.000852 S/cm, Supplementary Fig. 16, detailed calculations in Methods). Furthermore, the straight R-TiO$_2$/ZnWO$_4$ nanofibers with continuously interconnected structure can enhance the conductivity by 14.3 times compared to the separated nanofibers (Fig. 4b), further emphasizing the importance of interconnected fibrous mat structure to PEC application.

The significantly enhanced light absorption and electrical conductivity pave the way to PEC application. The in situ solid-state EPR spectra under light irradiation further proved the improved performance of R-TiO$_2$/ZnWO$_4$ nanofibers in generating photoexcited charges. EPR signal at *g* = 2.003 of TiO$_2$/ZnWO$_4$ (Supplementary Fig. 5 and 17) was observed and attributed to the spin electrons in CB. Due to the photoexcitation process, the signal's intensity increased by 1.31 times under light irradiation. Similarly, R-TiO$_2$/ZnWO$_4$ nanofiber also exhibited such an EPR signal but with an enhanced intensity under light irradiation. Notably, the enhancement of this EPR signal for R-TiO$_2$/ZnWO$_4$ nanofiber was elevated to 1.48 times. This suggested that OVs in R-TiO$_2$/ZnWO$_4$ nanofiber could receive photoinduced electrons[36], resulting in the reduced charge recombination and increased spin-electron concentration in CB[25].

Upon collecting the key factors for PEC application, we then examined the central part of our design–triphase diffusion. R-TiO$_2$/ZnWO$_4$ fibrous mat possesses multi-level porous nanochannels, offering affinity with electrolyte. To further expedite the access of CH$_4$ to photoanode surface, the R-TiO$_2$/ZnWO$_4$ fibrous mat was combined

with the $CH_4$-affinity poly(tetrafluoroethylene) (PTFE) with abundant gas diffusion pores (Supplementary Fig. 18). The closely combined interface between R-$TiO_2$/$ZnWO_4$ and PTFE ensured the formation of amphiprotic and porous triphasic interface as predesigned in Fig. 1a. As a result, this structure of triphasic interface will enable quick $CH_4$ diffusion to R-$TiO_2$/$ZnWO_4$ and prolonged adhesion of $CH_4$ reactant bubbles on active sites simultaneously, as schematically illustrated in Fig. 4c. We measured the contact angle of $CH_4$ gas bubbles in the electrolyte by a well-designed experimental setup (details can be found in "Methods")[37]. In the optical observation (Fig. 4c), the PTFE immediately adsorbed the $CH_4$ bubbles upon close approach and transferred them into the inner part toward R-$TiO_2$/$ZnWO_4$ active sites. This adsorption process took place so quickly within 10 ms that the detailed performance must be captured by a high-speed camera (1000 frames/s). The gas layer built by PTFE ensures the rapid $CH_4$ gas diffusion into the void nanopores of R-$TiO_2$/$ZnWO_4$.

Despite the commonly used PTFE layer, our R-$TiO_2$/$ZnWO_4$ fibrous behaves very differently from other reported advanced gas diffusion electrode. Our fibrous mat can adhere $CH_4$ gas for a prolonged time with small bubble size while simultaneously contacting aqueous electrolyte well. Figure 4c compared the gas adhesion time on R-$TiO_2$/$ZnWO_4$ fibrous interwoven mat, broken nanofibers and nanoparticles. The $CH_4$ gas was adhered on R-$TiO_2$/$ZnWO_4$ interwoven fibrous mat with longer time (3.2 s, Supplementary Fig. 19 and Fig. 4c), three and five times that of broken nanofibers and nanoparticles, respectively. As illustrated in Fig. 4c, the interwoven fibrous structure with micropores can enhance capillary force to stabilize gas bubble with more supporting walls as compared to other configurations. This prolonged $CH_4$ adhesion will make the reactant conversion more efficient and sufficient. In the meantime, the inherent hydrophilic property of R-$TiO_2$/$ZnWO_4$ nanofibers adds inimitable merits to overall performance. The hydrophilic surface will enable high ionic conductivity and favorable proton availability in PEC by facilitating timely exchange with electrolyte ions[5], and can preserve active sites from shielding by suppressing growth and accumulation of gas bubbles in a long-term reaction[38,39].

## PEC $CH_4$ conversion by hierarchical fibrous mat in flow cells

Our designed R-$TiO_2$/$ZnWO_4$ fibrous mat has been featured with superior abilities in free-standing mechanical stability, broad light absorption, suppressed charge recombination, high electrical conductivity, quick mass diffusion and harnessed triphasic interfaces, which endows it with the application as a diffusion photoanode in a continuous and rapid flow cell for $CH_4$ conversion. Figure 5a illustrates the PEC flow system by cycling $CH_4$ and electrolyte flow, which separates the two cells for gas and liquid with fibrous mat sandwiched inside. It turns out that photoelectric response, the first step in PEC $CH_4$ conversion, can be enhanced by our fibrous mat. Even in a conventional batch cell, the photocurrent of R-$TiO_2$/$ZnWO_4$ fibrous mat was increased by 55% as compared to pristine $TiO_2$/$ZnWO_4$. Under the flow conditions, the photocurrent (11 mA/g) was further improved by the fibrous mat (Fig. 5b), 28 times higher than that of the batch reaction at the same applied potential.

Based on the improved PEC current, value-added products, mainly as acetic acid ($CH_3COOH$), were continuously yielded from $CH_4$ conversion at high rate and mass activity (Fig. 5c and Supplementary Fig. 20). The $CH_3COOH$ production rate reached 9.3 mmol/g (normalized to catalysts mass) under an optimal potential of 0.55 V (vs. RHE) for 20 h, with a high Faradic efficiency on $CH_3COOH$ (90.4%), which was superior to the recent advanced reports on $CH_4$ conversion at the mass activity and efficiency[40–43]. The high mass activity is highly beneficial to the design of light-weighted, portable and highly integrated PEC $CH_4$ conversion cells in future industry. At lower applied potential (below 0.75 V, vs. RHE), $CH_4$ can be converted to both high-valued $CH_3COOH$ and formic acid (HCOOH) with a smaller amount of

$CO_2$ (Supplementary Fig. 20), owing to the reduced overoxidation. As calculated for electrocatalysis, the energy efficiency for $CH_3COOH$ ($EE_{CH_3COOH}$) is 114% and 195% by the fibrous mat in the flow cell for 2 h and 20 h, respectively, confirming the important contribution of light irradiation to this effective reaction. To gain insight into the reaction pathway of PEC $CH_4$ conversion on R-$TiO_2$/$ZnWO_4$, EPR measurement was performed under the same electrolyte and light irradiation reaction conditions while 5, 5-dimethyl-1-pyrroline N-oxide (DMPO) was used as a spin-electron trapping agent. As shown in Supplementary Fig. 21, the presence of ·OOH radicals in the reaction system under light irradiation indicated that the $CH_4$ oxidation was triggered by radicals. A stronger intensity of DMPO-OOH was observed for R-$TiO_2$/$ZnWO_4$, suggesting that the production of ·OOH radicals was enhanced by the addition of OVs. OVs acted as electron acceptor to generate ·OOH radical, facilitating $CH_4$ activation[44]. In acidic condition, the mild ·OOH radicals as the main active species can couple with the methyl radical (·$CH_3$) converted from $CH_4$, which leads to the further upgrading into HCOOH and $CH_3COOH$ without further oxidation of oxygenates[45].

Notably, the selectivity of $C_2$ product and the production rate of valued chemicals both can be remarkably improved by the designed diffusion fibrous mat and flow cell configuration. In particular, the $C_2$ product (referred to $CH_3COOH$ here) selectivity in flow reaction can reach up to 90% even after a long-time running under high flux of photons (applied potential: 0.55 V vs. RHE; light density: 4 Sun; running time: 20 h), achieving a 400% improvement over the selectivity in a batch reaction (Fig. 5d). Simultaneously, the production rate of $CH_3COOH$ in flow reaction by our mat with predesigned interconnected structure was 16.6 times higher than that in batch design (Fig. 5e). Moreover, even under the same batch conditions, the production rate can increase nearly 80% by fibrous mat with interconnected pore structures as compared to the fluorine doped tin oxide (FTO) electrode loaded with dispersed nanofibers (Fig. 5e), indicating the vital role of mass transfer within the photoanode.

To investigate the effect of dislocations on the thermodynamics of surface catalytic reactions, the PEC performance of R-$TiO_2$/$ZnWO_4$ samples with and without dislocations (see details in "Methods" and Supplementary Fig. 22) were tested in the same batch reactions without affecting kinetics by mass transfer. Under the potential of 0.75 V (vs. RHE) and the 1 Sun light irradiation for 2 h, the R-$TiO_2$/$ZnWO_4$ catalysts without dislocations showed a $CH_3COOH$ production rate of 7.19 μmol/g and a $CO_2$ production rate of 11.3 μmol/g, which were similar to those with dislocations ($CH_3COOH$ production rate of 7.67 μmol/g and $CO_2$ production rate of 13.6 μmol/g). Both the samples showed the same selectivity on liquid (referred to as $CH_3COOH$, 100%) and gas products (referred to as $CO_2$, 100%) at this test condition. Hence, the dislocations on the boundaries in R-$TiO_2$/$ZnWO_4$ had little thermodynamics influence on the activity and selectivity of the overall PEC reactions. This structure in nanofibers mainly enhanced the mechanical properties of interwoven catalyst mats, which laid the roots for remarkably promoting the kinetics performance by mass transfer under the high-speed fluid flowing during the PEC flow reactions.

We further examined the effect of flow rates on PEC flow reactions. With the increase of the input $CH_4$ flow rate, the diffusion rate of $CH_4$ increased within the fibrous interwoven pores on mat (Supplementary Fig. 23). With the $CH_4$ flow rate of 50 mL/min, the production rates of $CH_3COOH$ and HCOOH reached 1.86 and 1.20 mmol/g, respectively (7.38 and 6.25 times those at 25 mL/min), with a $C_2$ product selectivity of 53.9% (referred to as $CH_3COOH$, 1.26 times that at 25 mL/min). When the $CH_4$ flow rate was increased to 100 mL/min, the $CH_3COOH$ production rate reached 3.46 mmol/g (13.6 times that at 25 mL/min), with a $C_2$ product selectivity of 79.7% (1.86 times that at 25 mL/min). These results further confirmed the significant improvement on the surface catalytic reactions in PEC by enhancing mass transfer kinetics. Based on the free radical activation mechanism

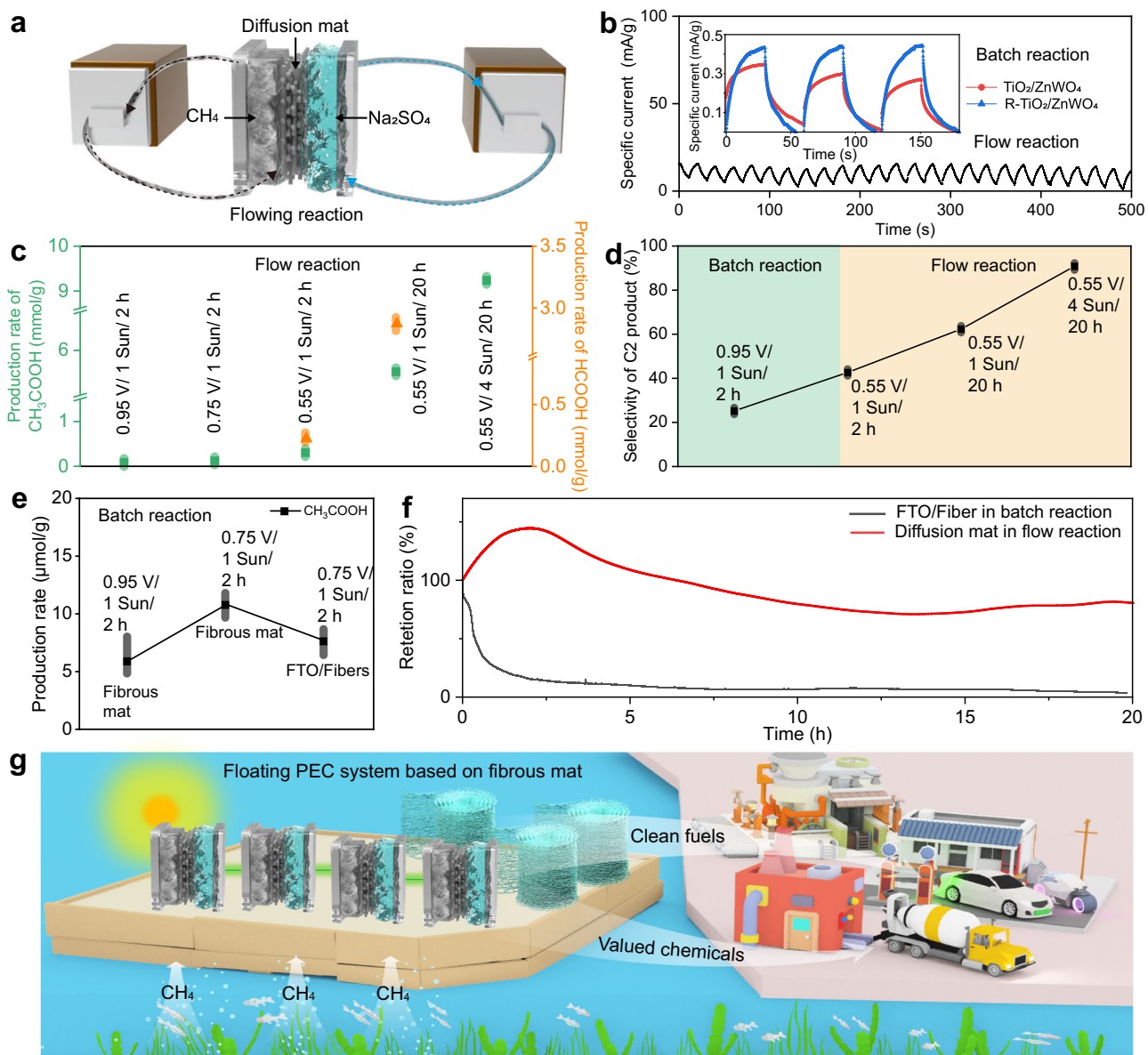

**Fig. 5 | PEC CH4 conversion performance of fibrous photoanode in a flow cell.**
**a** Schematic illustration of the flow system design with fibrous diffusion photoanode. **b** Current−time (I−t) curves of TiO₂/ZnWO₄ and R-TiO₂/ZnWO₄ photoanode in conventional batch cell, as well as I−t curve of R-TiO₂/ZnWO₄ electrode in flow cell. **c** Production rate of CH₄ conversion in flow cell by R-TiO₂/ZnWO₄ electrode. All the potential (V) was versus RHE. **d** Selectivity of C₂ product in batch and flow cells by R-TiO₂/ZnWO₄ photoanode. Comparison on production rate (**e**) and

specific current stability (**f**) of CH₄ conversion in flow cell by R-TiO₂/ZnWO₄ free-standing fibrous photoanode and in batch cell by FTO photoanode loaded with nanofibers. The stability tests were all conducted at 0.75 V vs. RHE. The error bars represent standard deviations in (**c**−**e**). **g** Schematic illustration of the floating PEC system based on our triphase diffusion photoanodes for locally converting CH₄ gas into value-added products solely driven by sun light.

mentioned before, the faster flow promoted the migration of methyl radicals for coupling[12], thereby enhancing the CH₄ conversion as well as the reaction between methyl radicals and HCOOH into CH₃COOH products. Moreover, the shorter contact time reduced the possibility of unexpected overoxidation reactions to other products such as CO₂[12,46], thus improving the C₂ product selectivity.

Furthermore, thanks to the free-standing and mechanically strong structure, the stability of CH₄ conversion was significantly enhanced. As shown in Fig. 5f, the specific current of flow reaction by our diffusion mat stayed stable even under a rapid fluid shock condition for 20 h while the retention ratio of FTO with loaded nanofibers was below 20%, at the same applied potential (0.75 V vs. RHE) under light irradiation (1 Sun). Apparently, the fibrous mat remained intact after stability test (Supplementary Fig. 24). Notably, our diffusion fibrous mat

can work stably for at least 100 h by controlling working conditions (Supplementary Fig. 25), which holds great potential in practical applications. These results clearly demonstrate that our diffusion mat photoanode can enhance the production rate, selectivity and stability of valued chemicals from continuous PEC CH₄ conversion in flow cells.

It is anticipated that the flexible and light-weight semiconductor mat can be folded into free-standing photoelectrodes with various portable or light absorption-favorable structures in the future, directly catalyzing CH₄ gas at the local sea where CH₄ is extracted. This strategy will allow for constructing of a floating solar-to-valued products PEC system, not limited to CH₄ conversion. Specifically for CH₄ conversion, clean fuels and chemicals can be continuously produced by in situ converting the CH₄ gas pumped from seafloor (Fig. 5g) for supporting the civil life.

## Discussion

To address the mass transfer limitations in PEC gas/liquid conversion system, we have developed a first reported triphase diffusion photoelectrode for continuous flow cell, based on free-standing hierarchical fibrous mats. Our fabrication approach induced a high density (-10$^{10}$ mm$^{-2}$) of dislocations among the refined TiO$_2$/ZnWO$_4$ nanocrystals in 1D nanofiber, and also overcame the intractable inherent brittleness of PEC semiconductor. The fabricated TiO$_2$/ZnWO$_4$ fibrous mat possessed an ultrahigh specific strength (over 90,000 MPa/cm$^3$/g), and was free of deformations without cracking even when high-speed (100 mL/min) gas or aqueous electrolyte flowed through continuously. Such a strong fibrous skeleton can be retained in any further modifications, such as creation of OV active sites along the pore surfaces of nanofibers. Leveraging the skeleton, gas/liquid amphoteric interfaces can be constructed to simultaneously ensure efficient diffusion of both gaseous reagent and aqueous electrolyte, enabling the design of triphase diffusion photoelectrodes. As a proof of concept, this design has been successfully implemented in continuous CH$_4$ conversion in PEC flow cells. As a result, the production rate and C$_2$ product selectivity were both increased by 16.6 and 4.0 times as compared to batch reactions. The production rate of 9.3 mmol/g for 20 h, together with selectivity of 90% toward acetic acid at applied potential of 0.55 V (vs. RHE), exceeds the state-of-the-art reports on PEC CH$_4$ conversion. The designed triphase diffusion photoanode can stably work for at least 100 h. This work that provides fresh insights into nanostructured multifunctional materials with enhanced mass transfer in multiphase reactions, is expected to offer guidelines for the rational design of free-standing and flexible diffusion photoelectrodes toward scale-up solar-driven chemical transformations in the future.

## Methods

### MD simulation

All the MD simulations were performed with the LAMMPS package[47]. Periodic boundary conditions were applied in all three directions. Nose–Hoover thermostat was employed and the leapfrog Verlet algorithm was used to integrate the Newton's equations of motion with a timestep of 1 fs. The water molecules were represented by TIP4P/2005 rigid models[48], which can reproduce accurately the experimental values of both the viscosity and the diffusion coefficient[49,50]. The methane molecules and other interactions were described by Lennard–Jones potentials ($V$):

$$V(\mathbf{r}) = 4\varepsilon[(\frac{\sigma}{\mathbf{r}})^{12} - (\frac{\sigma}{\mathbf{r}})^{6}] \tag{1}$$

where $\mathbf{r}$ is the distance between particles, and $\varepsilon$ and $\sigma$ are energy and length parameters, respectively[51]. For methane molecules, the energy and length parameters were 0.293 kcal/mol and 0.3730 nm, respectively. The interaction of hydrogen atoms among water and methane molecules was neglected. The solids including piston, hydrophobic and hydrophilic walls were described by Lennard–Jones potentials. The energy and length parameters were 7.951 kcal/mol and 0.2644 nm, respectively. In addition, a spring force was applied to each solid and piston particle, tethering it to its initial position in order to prevent melting. Lorentz–Berthelot combining rules were used to derive interaction parameters for different types of atoms:

$$\varepsilon_{ij} = k\sqrt{\varepsilon_{ii}\varepsilon_{jj}} \tag{2}$$

$$\sigma_{ij} = \frac{(\sigma_{ii} + \sigma_{jj})}{2} \tag{3}$$

For water–methane interactions, a slight modification for $k = 1.07$ can describe the water–methane interface behavior with high accuracy[52]. For different wetting behaviors of wall–water interactions,

the energy parameters were adjusted by a test simulation of contact angles[53]. In detail, 0.5 and 0.2 kcal/mol were adopted to model hydrophilic and hydrophobic walls, respectively. For hydrophilic wall–methane interaction, 0.5 kcal/mol was chosen to model a strong adsorption of gas on the catalyst region. To reduce the effect of pistons, the energy parameters of water and methane with pistons was just 0.05 kcal/mol.

A composite nanochannel was used to connect two reservoirs. The left and right reservoirs were filled by water and methane molecules, respectively. The left and right parts of the composite nanopore were hydrophilic and hydrophobic, respectively. Pistons at two ends were used to enclose the liquid and gas, and to avoid the possible liquid–gas interaction and mixing induced by periodic boundary conditions. The system was firstly run under $NVT$ (particle number $N$, volume $V$ and temperature $T$ are constant) ensemble at $T = 293$ K for 1,000,000 timesteps. It was long enough for water to enter the hydrophilic part of the nanochannel and to reach a steady state. Then, 2,000,000 timesteps were continued to obtain related statistical data. Three independent simulations with different initial velocities were performed and the results were averaged to reduce statistical uncertainties.

### CFD simulation

All CFD simulations were carried out with a commercial software (ANSYS FLUENT 2022 R1). As shown in the simulation model (Supplementary Fig. 1a–c), two fluid reservoirs were connected by the fiber region. In both simulations and experiments, the fluid input pressure and flow rate were 2.38 MPa and 0.074 m/s, respectively. We measured the fluid flowing pressure using high precision digital gauges (YK-100B, Schlocker Instrument Technology Co., Ltd., China) connected to the fluid delivery pipelines. Besides, to simulate the crossing of the fiber materials, 10 layers were used. Mat layers were created as shown in Supplementary Fig. 1b. Layers were formed by successively rotating one basics mat layer 20° around the center. In fluid dynamics simulations, the number of fibrous mat layers should be at least ten, to ensure that the fibrous catalysts are not overwhelmed by the electrolyte fluid and can perform the turbulence flowing behavior at fibrous pores. With the increase of layer numbers, the average fluid flow rate within mat decreased (Supplementary Fig. 26). For this reason, we experimentally controlled the small thickness of catalytic mat and increased fibrous mechanical strength to ensure the structural stability of thin mat during high-speed fluid flow. Different parts were connected via interface function, and structured meshes were constructed for each part. Volume of fluid (VOF) model was employed to simulate the flow of the incompressible water and methane[54]. The surface tension force modeling was enabled and the method of continuum surface force was used[55], for which surface tension was constant along the water–methane interface and only the forces normal to the interface were considered. The flow was assumed to be laminar due to the low Reynolds (Re) number in the realistic experiments (Re: -500).

### Materials

Titanium tetraisopropoxide (97%), polyvinylpyrrolidone ($M_w \approx 1.3 \times 10^6$) and dimethyl sulfoxide (99.9 atom% D, contains 0.03% TMS) were purchased from Sigma-Aldrich. Ethanol (anhydrous, 94–96%) was purchased from Alfa Aesar. Ammonium tungstate hydrate and deuterium oxide (D$_2$O, 99.9 atom% D) were obtained from Macklin Chemical Reagent Co., Ltd. The CH$_4$ gas (99.9%) was obtained from Nanjing Changyuan Industrial Gases Co., Ltd. Other chemicals were all obtained from Sinopharm Chemical Reagent Co., Ltd. All chemicals were used as received. The water used in all experiments was filtered through a Millipore filtration system with resistivity above 18 MΩ·cm.

### Fabrication of TiO$_2$/ZnWO$_4$ nanofiber mat

The ZIF-8 was synthesized by following procedure: 6.78 g of Zn(NO$_3$)$_2$·6H$_2$O was dissolved into 200 mL methanol as solution A, and

7.87 g of 2-methylimidazole was dissolved into 200 mL methanol as solution B. Then, solution A was rapidly added into solution B within 30 s by a separating-funnel and magnetically stirred for 1 h. ZIF-8 powders were obtained after centrifugation (8000 rpm, 5 min) and washing of products with methanol for 3 times, along with the vacuum drying at 50 °C for 40 h[56]. Typically, polyvinylpyrrolidone (0.6 g, $M_w = 1.3 \times 10^6$), ammonium tungstate hydrate (0.08 g), zeolite imidazole frameworks (ZIF-8, 0.13 g) and titanium isopropoxide (2.5 mL) were mixed with ethanol (4.5 mL), dimethylformamide (3.0 mL) and acetic acid (3.0 mL) to form a yellow solution after magnetic stirring for 12 h. Afterward, the solution was transferred into two 10 mL syringes capped with 21 G needles in an electrospinning setup with a voltage of 20 kV. Meanwhile, the precursor solutions were squeezed at a controllable rate of 1.5 mL/h at 25 °C within an environment with 40% RH. The needle-to-collector distance was 12.5 cm. The collected $TiO_2/ZnWO_4$ precursor was calcinated at 600 °C for 2 h with a temperature ramping rate of 2 °C/min in air atmosphere. By adjusting the electrospinning time and collector, the transparent mat was obtained. The $TiO_2/ZnWO_4$ fibrous mat was sprayed with a 1 mol/L $NaHB_4$ solution to obtain $R-TiO_2/ZnWO_4$ through reduction, then sprayed with water 5 times to remove the $NaHB_4$ residue on the surface and dried in a vacuum dryer. For the application of flow reaction, $R-TiO_2/ZnWO_4$ was combined with porous PTFE mat by a quick hot press (80 °C, 15 s) for further use. The transparent mats were collected by a stainless-steel mesh with both the self-standing and light-transmission microstructures, as shown in Supplementary Fig. 14.

## Characterization

High-resolution TEM images and in situ mechanical test were taken on a FEI Titan 80-300 spherical aberration corrected transmission electron microscope at 300 kV. The in situ mechanical test on TEM was conducted on a mechanical sample rod with W probe. The high-angle annular dark-field (HAADF)-STEM images and EDS mapping profiles were measured on the Talos F200X field-emission transmission electron microscope operated at 200 kV. EELS mapping data were collected on the FEI Titan3 G2 60−300 double spherical aberration correction electron microscope at 300 kV. GPA analysis were conducted on Digital Micrograph Software based on the HRTEM images. Nanofiber slices analyzed in Fig. 2b were cut by focused ion beam scanning electron microscopy (FIB-SEM, Helios 5 CX) along the radial direction of a long nanofiber. SEM images were obtained from FEI Inspect F50. The mechanical tests were performed on a universal tensile tester (5940, Instron Co., Ltd, America). The samples in tensile and compression tests were all loaded to the same strain rate at 1.5 mm/min. EPR spectra were obtained on the JEOL JES-FA200 spectrometer equipped in dark or with a Xenon lamp with light intensity of 1 Sun as the illumination source. $Na_2SO_4$ electrolyte with pH value of 2 was used as solution in EPR test with 1 Sun light irradiation for simulating the PEC reaction condition. Powder XRD patterns were recorded by using the Ultima IV Philips combined multifunctional horizontal X-ray diffractometer with Cu-Kα radiation (λ = 1.54178 Å). XPS spectra were collected on XPS-2 multifunctional photoelectron spectrometer.

The DIC experiment was carried out in a real flow cell with separate flows of $CH_4$ and electrolyte across the same mat. Mats were pre-sprayed with paint speckles. The mist spray of paint was attached to the specimen surface, forming the random speckles with diameter of 50−150 μm. A binocular camera system was set up and calibrated with a circle calibration board to measure the 3D deformation of the specimen surface non-destructively. The region of interest (ROI) accounted for approximately 60 × 60 pixels in the image. The setup of cells for DIC test were shown in Supplementary Fig. 11. In the light window position of the PEC reactor, we pre-flattened the mat with adhesive tapes and firmly fixed it with a rigid stainless-steel sheet. The flow rate was set by peristaltic pump. Both the gas and liquid flows were kept the same as those of real continuous flow PEC reactions.

In conductivity test of nanofiber, single nanofiber was aligned between two testing probes to form an approximate parallel circuit. The aligned $TiO_2/ZnWO_4$ nanofibers were collected on a glass by high-speed (1800 rpm) rotating collector for 20 s. Then, the fibers were reduced by $NaHB_4$ via the above spraying method. Aligned $TiO_2/ZnWO_4$ nanofibers and aligned $R-TiO_2/ZnWO_4$ nanofibers were prepared to a conductivity testing electrode, followed by chemical vapor deposition of Au (10 mA, 90 s) under a designed mask. The Au coating area served as electrode collector, while the aligned nanofibers between Au areas formed parallel circuits, as shown in Supplementary Fig. 15. The current-voltage curves were recorded by conductivity probe station (HCP621G-PM, INSTEC) with Keithley 6517B resistance meter. The probes were set on Au areas. The conductivity was calculated by the following Eq. (4):

$$\sigma = \frac{I}{AR} \tag{4}$$

where $\sigma$ is the conductivity (S/m), $A$ is the cross-sectional area of nanofiber, $I$ is the length of nanofiber, and $R$ is the resistance measured from the Keithley resistance meter.

The contact angles for $CH_4$ gas bubbles in $NaSO_4$ electrolyte (pH = 2) on PTFE and $R-TiO_2/ZnWO_4$ surfaces were measured in the liquid electrolyte by a captive bubble method. In detail, the $CH_4$ gas was prestored in the needle tube with switches and then carefully pumped around the solid surface of the PTFE and $R-TiO_2/ZnWO_4$ surfaces on FTO glass. Then, we observed the $CH_4$ gas bubble evolution on photoanode by optical microscopy for investigating their gas contact angles[37]. The interval-time photographs of methane retention on $R-TiO_2/ZnWO_4$ fibrous mat were shown in Supplementary Fig. 19. The timing began when methane bubbles contacted with the mat's surface.

## PEC $CH_4$ conversion measurements

PEC batch and flow measurements were carried out on an electrochemical workstation (CHI 660E) in a sealed H-type cell and a sealed flow cell (details shown in Supplementary Fig. 27), respectively, with Nafion 117 proton exchange membrane separated. All PEC tests were conducted at room temperature and atmospheric pressure. The simulated solar illumination was obtained from a 300 W Xenon lamp (Beijing AuLight, CEL-HXF300-T3) with output light spectra from 300 to 2500 nm at a power density of 100 mW/cm². The light intensity was adjusted by an optical power meter (S314C, THORLABS) with an integrated broadband sensor, and the intensity meter was set at the same height as the photoanode. In all measurements, the photoanodes were front-illuminated and immersed 1 cm² (average catalyst mass of 3.57 mg on FTO glass in batch reactions and 0.83 mg without additional substrates in flow reactions) in electrolyte as the working electrode, while Ag/AgCl electrode and Pt sheet (1 cm²) were used as the reference and counter electrode, respectively. The electrolyte contained 0.1 M $Na_2SO_4$ aqueous solution with pH adjusted to 2 using 0.5 M $H_2SO_4$. $CH_4$ gas was purged for 30 min before reaction. The average oxygen content in PEC electrolyte after pre-pouring methane gas was 1.39 mg/L, which was tested by a dissolved oxygen meter (JPSJ-605, INESA Scientific Instrument Co., Ltd, China). $CH_4$ gas and electrolyte liquid were cycled by peristaltic pumps with a fluid rate of 25 mL/min in flow reaction. The gas products from PEC $CH_4$ conversion were analyzed using a gas chromatograph (GC, 7890B, Ar carrier, Agilent) equipped with a thermal conductivity detector (TCD) and a flame ionization detector (FID). The GC was also equipped with a methanation reactor[43]. The liquid products were quantified using nuclear magnetic resonance (NMR) spectroscopy. ¹H NMR spectra were collected on Bruker AVANCE AV III 400 spectrometer in 10% $D_2O$ using the water suppression mode, with dimethyl sulfoxide as the internal standard. The ¹H NMR spectra of liquid products in different

flow reaction conditions were shown in Supplementary Fig. 28. All measurements were calibrated with iR-compensation.

The R-TiO$_2$/ZnWO$_4$ counterpart samples were prepared through electrospray under the same electric field and calcination treatment as the original electrospun nanofibers, but without the growth confinement in ultrathin fibers (Supplementary Fig. 22a). The precursor in electrospray was obtained via diluting the electrospinning precursor by ethanol solution with a volume dilution ratio of 1: 2, and then was squeezed at a rate of 3 mL/h through a 19 G needle. The resulting counterparts had the same components and exposed crystal planes as the nanofibers, but lacked the dislocation structures on boundaries (Supplementary Fig. 22b, c). These contrast samples were coated on FTO glasses for PEC batch reactions. The $^1$H NMR spectra of liquid products in batch reaction conditions by R-TiO$_2$/ZnWO$_4$ with and without dislocations were shown in Supplementary Fig. 29. The CO$_2$ production rate by the R-TiO$_2$/ZnWO$_4$ mat in flow reactions with CH$_4$ flow rates of 50 and 100 mL/min, was 0.532 and 0.877 mmol/g, respectively. Contrast experiments with increasing CH$_4$ flow rates were carried out in the flow reactor for 2 hours at a potential of 0.55 V (vs. RHE) and a light intensity of 1 Sun. The $^1$H NMR spectra of liquid products in flow reactions by R-TiO$_2$/ZnWO$_4$ mat with increasing CH$_4$ flow rates were shown in Supplementary Fig. 30.

Electrode potentials were rescaled to the RHE reference by the following equation, E (vs. RHE) = E (vs. Ag/AgCl) + 0.196 V + 0.059 × pH. The Faradaic efficiency (FE) at applied potential of 0.55 V (vs. RHE) was calculated on the basic of the following equation (5):

$$FE = \frac{Q_X}{Q_{total}} = \frac{n_X N_X F}{Q_{total}} \qquad (5)$$

where $Q_x$ and $Q_{total}$ was the charge passed into product x and totally passed charge (C) during PEC CH$_4$ oxidation, $n_x$ represents the electron transfer number of product x, $N_x$ was the product amount (mol) of x measured by GC or NMR, and F was the Faraday constant (96485 C/mol).

The electrocatalysis energy efficiencies (EE) were calculated on the basic of the following equation (6):

$$EE = \frac{E^\theta}{E_{applied}} \times FE_{CH_3COOH} \qquad (6)$$

where $E^\ominus$ is the thermodynamic potential for the acetic acid formation (1.19 V vs. RHE, at 25 °C), $E_{applied}$ represents the potential applied during the PEC CH$_4$ conversion. FE$_{CH3COOH}$ is the Faradaic efficiency to acetic acid.

## Data availability

The authors declare that all data supporting the findings of this study are available in the article and its Supplementary Information. Source data are provided as a Source Data file and have also been deposited in figshare under accession code https://doi.org/10.6084/m9.figshare.22569121[57]. Source data are provided with this paper.

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

## Acknowledgements

This work was financially supported by the National Natural Science Foundation of China (21975042, 22232003, 12174050, 21725102), National Key Research and Development Program of China (No. 2020YFC1511902, 2022YFA1505700), the Project of Six Talents Climax Foundation of Jiangsu (XCL-082), and the Priority Academic Program Development of Jiangsu Higher Education Institutions. The authors thank the support from Professor Shuai Dong in School of Physics in Southeast University for the conductivity test of nanofiber, as well as the TEM characterization by Mingyu Tang in School of Chemistry and Chemical Engineering in Southeast University.

## Author contributions

Y.D., Y.X. and X.M. conceived the idea. X.M. and C.Z. carried out the experiments and performed the measurements. X.W. performed the theoretical calculations. K.Y. and L.S. supported the TEM studies. M.Z. performed the fibrous conductivity studies. Z.L. supported the PEC products studies. L.G. and X.S. supported the DIC tests. Y.D., Y.X., Y.S. and R.L. discussed the research. X.M., X.W., Y.D. and Y.X. wrote the manuscript.

## Competing interests

The authors declare no competing interests.
