## [Peer Review File · Nature Communications]

Hierarchical triphase diffusion photoelectrodes for photoelectrochemical gas/liquid flow conversionREVIEWER COMMENTS

Reviewer #1 (Remarks to the Author):

The sluggish mass and electron transfer on gas/photoelectrode/electrolyte triphase interfaces are a bottleneck which largely limits the PEC practical applications for a long time. Excitingly, this article reports a first design of hierarchical triphase diffusion photoelectrodes for PEC gas/liquid flow conversion, resulting in largely improved mass transfer on triphase interfaces. The work can attract wide attention on designing mass-transfer-accelerating materials in more multiple-phase reactions and is vital to scale-up PEC chemical industry. Although the work is valuable, the authors still should resolve some issues in minor revision before being accepted for publication:

(1) The water and methane density results are largely related to the input pressure setting in fluid simulations. Is the pressure similar to the real experimental conditions? Authors should provide the experimental test detail on measuring fluid flowing pressure in devices and more setting details in simulations.

(2) Except for the pore structures, is the fluid flowing behavior related to the fibrous mat layer numbers? Provide some relationship discussions on simulation results.

(3) Authors should explain more on the forming mechanism of high-density dislocations and grain boundary stress. Is this behavior related to the ultra-thin structure of nanofibers? Are there any different key points compared to other electrospun nanofiber reports?

(4) Authors should provide more mechanical tests on R-TiO₂/ZnWO₄. Also, considering that the real pressure during fluid flowing can be in both tensile and compression ways, some compression strength tests should be supplemented. What about the unit rupture work?

(5) In DIC test, considering the flexibility of mat, how the authors pretension the mat in real light window position? The details of method should be added.

(6) The analysis on bandgaps of samples in Fig. 4a should be discussed.

(7) Please provide the microstructure of transparent mat in supporting information. Is it similar to the non-transparent one?

(8) Authors should supplement the interval-time photographs of methane retention on mat.

(9) What is the average oxygen content in the PEC electrolyte in this work?

Reviewer #2 (Remarks to the Author):

In this manuscript, the authors fabricated the hierarchical triphase diffusion photoelectrodes for building a continuous flow cell for the photooxidation of CH₄ to acetic acids or other products. The photoelectrode contained a structure of interwoven nanofibers of TiO₂/ZnWO₄ with introduced oxygen vacancies. Such a structure can enhance mechanical specific strength and flexibility for facilitating mass transfer of chemicals, such as reactants, solvent, and products. Meanwhile, the photoabsorption and photocurrent are also increased. Due to these effects, significantly improved catalytic activity and selectivity was observed by the photoelectrodes for the conversion of CH₄ to value-added chemicals. Comprehensive computational and experimental data were provided to support the argument. This work is surely interesting to the research communication of photocatalytic conversion of CH₄. I would suggest the manuscript to be accepted in the journal of Nature Communications, with a minor revision.

1. What is the "high surface energy side" in the design TiO₂/ZnWO₄ structure?

2. The authors stated that creating OVs does not influence much the mechanical properties of TiO₂/ZnWO₄. "In our case, benefiting from interconnected pore channels, the reductant was mostly transferred to and reacted with the nanofiber surface and the pore walls, thus remaining a rather intact and strong fibrous skeleton." What are the experimental data to support the statement, by comparing TiO₂/ZnWO₄ and R- TiO₂/ZnWO₄?

3. In addition to influencing the reaction kinetics by mass transfer, how may the unique structure of TiO₂/ZnWO₄ nanofibers (e.g., dislocation on the boundary influence the thermodynamics of surface catalytic reactions, e.g., changing the surface adsorption of CH₄, H₂O, or products? Does

the surface adsorption energy of CH₄, H₂O, or products influence the activities or selectivity of the overall photocatalytic reactions here? Why or why not?

4. In this photoelectrode system, how can the increase of CH₄ flow rate influence the diffusion rate of CH₄ and/or the kinetics surface catalytic reactions?

5. What is the reason for the increased selectivity towards acetic acid in the flowing reaction, compared to the batch reaction.

Reviewer #1 (Remarks to the Author):

The sluggish mass and electron transfer on gas/photoelectrode/electrolyte triphase interfaces are a bottleneck which largely limits the PEC practical applications for a long time. Excitingly, this article reports a first design of hierarchical triphase diffusion photoelectrodes for PEC gas/liquid flow conversion, resulting in largely improved mass transfer on triphase interfaces. The work can attract wide attention on designing mass-transfer-accelerating materials in more multiple-phase reactions and is vital to scale-up PEC chemical industry. Although the work is valuable, the authors still should resolve some issues in minor revision before being accepted for publication:

We really appreciate the referee's highly positive evaluation for our work, and are grateful to the referee for his/her insightful suggestions to help us further improve the quality of our manuscript.

(1) The water and methane density results are largely related to the input pressure setting in fluid simulations. Is the pressure similar to the real experimental conditions? Authors should provide the experimental test detail on measuring fluid flowing pressure in devices and more setting details in simulations.

We thank the referee for his/her valuable suggestion. In both simulations and experiments, the fluid input pressure and flow rate were 2.38 MPa and 0.074 m/s, respectively. We measured the fluid flowing pressure using high precision digital gauges (YK-100B, Schlocker Instrument Technology Co., Ltd., China) connected to the fluid delivery pipelines. We have now included the related test and simulation details in Methods on Page 28 in the revised manuscript.

(2) Except for the pore structures, is the fluid flowing behavior related to the fibrous mat layer numbers? Provide some relationship discussions on simulation results.

We thank the referee for his/her insightful suggestion. In computational fluid dynamics simulations, the fluid flowing behavior is related to the fibrous mat layer numbers. The number of fibrous mat layers should be at least ten, to ensure that the fibrous catalysts are not overwhelmed by the electrolyte fluid and can perform the turbulence flowing behavior at fibrous pores. However, with the increase of layer numbers, the average fluid flow rate within mat decreases (the newly added Supplementary Fig. 26). As a result, in experiments, we carefully controlled the small thickness of catalytic mat and increased fibrous mechanical strength to ensure the structural stability of thin mat during high-speed fluid flow in flow cells. We have added the relationship between fibrous mat layer numbers and fluid flowing behavior (i.e., average flow rate) as Supplementary Fig. 26. The related discussions on simulation results have also been added on Page 29 in the revised manuscript.

(3) Authors should explain more on the forming mechanism of high-density dislocations and grain boundary stress. Is this behavior related to the ultra-thin structure of nanofibers? Are there any different key points compared to other electrospun nanofiber reports?

We thank the referee for his/her thoughtful suggestion. Compared to the common electrospun nanofibers, our fiber is ultrathin with an average diameter of 109 nm. TiO₂ and ZnWO₄ crystals are restricted in the small radial space of the ultrathin nanofiber during calcination, resulting in the refining of crystals and high slippage force among crystals. As a result, high-density dislocations were formed among the grain boundaries under the nonequilibrium condition at high temperature. We have added the related discussion and Ref. 21 and 22 on Page 11 and 36 in the revised manuscript.

(4) Authors should provide more mechanical tests on R-TiO₂/ZnWO₄. Also, considering that the real pressure during fluid flowing can be in both tensile and compression ways, some compression strength tests should be supplemented. What about the unit rupture work?

We thank the referee for his/her insightful suggestion. R-TiO₂/ZnWO₄ showed a tensile strength of 1.19 MPa and Young's modulus in tension of 62.2 MPa, as well as a tensile strain of 1.84%, indicating that reduction process had little influence on the mechanical properties of TiO₂/ZnWO₄. Because the real pressure during fluid flow can also be in compression mode, R-TiO₂/ZnWO₄ compression tests were carried out. The sample showed a compressive strength of 0.0761 MPa and Young's modulus in compression of 3.52 MPa, with a satisfying ductility strain of 9.34%. The unit rupture work of R-TiO₂/ZnWO₄ in compression was as high as 3100 J m⁻³ g⁻¹, outperforming previously reported advanced ceramic materials (*e.g.*, *Small* 2022, 18, 2201039; *Adv. Funct. Mater.* 2020, 30, 2005928; *ACS Nano* 2021, 15, 18354). The test results have been included in the newly added Supplementary Fig. 10. We have added the related discussions, test methods and Ref. 32-34 on Page 14, 30 and 36 in the revised manuscript, respectively.

(5) In DIC test, considering the flexibility of mat, how the authors pretension the mat in real light window position? The details of method should be added.

We thank the referee for his/her thoughtful suggestion. In the light window position of the PEC reactor, we pre-flattened the mat with adhesive tapes and firmly fixed it with a rigid stainless-steel sheet. The stainless-steel sheet structure is shown in Supplementary Fig. 11b. We have added the related details in Methods on Page 31 in the revised manuscript.

(6) The analysis on bandgaps of samples in Fig. 4a should be discussed.

We thank the referee for his/her valuable suggestion. Using the Tauc plot method, we calculated the bandgaps of the samples. TiO₂, TiO₂/ZnWO₄ and R-TiO₂/ZnWO₄ had direct bandgaps of

3.18, 2.51 and 2.40 eV, respectively. After integration and reduction, the bandgap of the sample decreased, which is beneficial to the broadening of the light absorption range of PEC catalysts. We have added the related discussion and Ref. 35 on Page 17 and 37 in the revised manuscript, respectively.

(7) Please provide the microstructure of transparent mat in supporting information. Is it similar to the non-transparent one?

We thank the referee for his/her valuable suggestion. The transparent mats collected by a stainless-steel mesh had both the self-standing and light-transmission microstructures. The microstructures have been shown in SEM images with a schematic functional illustration in the newly added Supplementary Fig. 14. We have also added the related discussion on Page 17 and 30 in the revised manuscript.

(8) Authors should supplement the interval-time photographs of methane retention on mat.

We thank the referee for his/her thoughtful suggestion. We have now included the interval-time (*i.e.*, 0, 0.8, 1.6, 3.2 and 3.21 s) images of methane bubbles on mat in the newly added Supplementary Fig. 19, for recording the methane retention (from 0 to 3.2 s) and releasing (at 3.21 s) behavior. The related discussion has been added on Page 19 and 32 in the revised manuscript.

(9) What is the average oxygen content in the PEC electrolyte in this work?

We thank the referee for his/her valuable suggestion. The average oxygen content in PEC electrolyte after pre-pouring methane gas was 1.39 mg/L, which was tested by a dissolved oxygen meter (JPSJ-605, INESA Scientific Instrument Co., Ltd., China). We have added the oxygen content in Methods on Page 33 of the revised manuscript.

Reviewer #2 (Remarks to the Author):

In this manuscript, the authors fabricated the hierarchical triphase diffusion photoelectrodes for building a continuous flow cell for the photooxidation of CH₄ to acetic acids or other products. The photoelectrode contained a structure of interwoven nanofibers of TiO₂/ZnWO₄ with introduced oxygen vacancies. Such a structure can enhance mechanical specific strength and flexibility for facilitating mass transfer of chemicals, such as reactants, solvent, and products. Meanwhile, the photoabsorption and photocurrent are also increased. Due to these effects, significantly improved catalytic activity and selectivity was observed by the photoelectrodes for the conversion of CH₄ to value-added chemicals. Comprehensive computational and experimental data were provided to support the argument. This work is surely interesting to the research communication of photocatalytic conversion of CH₄. I would suggest the manuscript to be accepted in the journal of Nature Communications, with a minor revision.

We really appreciate the referee's highly positive evaluation for our work, and are grateful to the referee for his/her insightful suggestions to help us further improve the quality of our manuscript.

1. What is the "high surface energy side" in the design TiO₂/ZnWO₄ structure?

We thank the referee for his/her valuable suggestion. The "high surface energy side" is the hydrophilic side, which generally results in good liquid wetting with a low contact angle. We have added the explanation on Page 7 in the revised manuscript.

2. The authors stated that creating OVs does not influence much the mechanical properties of TiO₂/ZnWO₄. "In our case, benefiting from interconnected pore channels, the reductant was mostly transferred to and reacted with the nanofiber surface and the pore walls, thus remaining a rather intact and strong fibrous skeleton." What are the experimental data to support the statement, by comparing TiO₂/ZnWO₄ and R-TiO₂/ZnWO₄?

We thank the referee for his/her insightful suggestion. R-TiO₂/ZnWO₄ had a tensile strength of 1.19 MPa, a Young's modulus in tension of 62.2 MPa and a tensile strain of 1.84%, indicating that creating OVs did not influence much the mechanical properties of TiO₂/ZnWO₄. We have added this experimental data to support the statement in the newly added Supplementary Fig. 10 as well as the related discussion and test methods on Page 14 and 30 in the revised manuscript, respectively.

3. In addition to influencing the reaction kinetics by mass transfer, how may the unique structure of TiO₂/ZnWO₄ nanofibers (e.g., dislocation on the boundary influence the thermodynamics of surface catalytic reactions, e.g., changing the surface adsorption of CH₄,

H₂O, or products? Does the surface adsorption energy of CH₄, H₂O, or products influence the activities or selectivity of the overall photocatalytic reactions here? Why or why not?

We thank the referee for his/her insightful suggestion. The R-TiO₂/ZnWO₄ counterpart samples were prepared through electrospray under the same electric field and calcination treatment as the original electrospun nanofibers, but without the growth confinement in ultrathin fibers (the newly added Supplementary Fig. 22a). The resulting counterparts had the same components and exposed crystal planes as the TiO₂/ZnWO₄ nanofibers, but lacked the unique dislocation structures on boundaries (the newly added Supplementary Fig. 22b and c). To investigate the effect of dislocations on the thermodynamics of surface catalytic reactions, R-TiO₂/ZnWO₄ samples with and without dislocations were coated on FTO glasses and their PEC performance were tested in the same batch reactions without affecting kinetics by mass transfer.

By R-TiO₂/ZnWO₄ catalysts, CH₄ was converted to high-valued CH₃COOH with a small amount of CO₂ under the potential of 0.75 V (vs. RHE) and the 1 Sun light irradiation for 2 h. The R-TiO₂/ZnWO₄ catalysts without dislocations showed a CH₃COOH production rate of 7.19 μmol/g (normalized to catalyst mass) and a CO₂ production rate of 11.3 μmol/g, which were similar to that with dislocations (CH₃COOH production rate of 7.67 μmol/g and CO₂ production rate of 13.6 μmol/g). Both the samples showed the same selectivity on liquid (referred to as CH₃COOH here, 100%) and gas (referred to as CO₂ here, 100%) products. Hence, the dislocations on the boundaries in R-TiO₂/ZnWO₄ had nearly no thermodynamics influence on the activity and selectivity of the overall PEC reactions. This unique structure in nanofibers mainly enhanced the mechanical properties of interwoven catalyst mats, which laid the roots for remarkably promoting the kinetics performance by mass transfer under the high-speed fluid flowing during the PEC flow reactions.

We have included the structural analysis of the counterpart samples without dislocations in the newly added Supplementary Fig. 22, as well as the ¹H NMR spectra of liquid products in batch reaction conditions by the R-TiO₂/ZnWO₄ samples with and without dislocations in the newly added Supplementary Fig. 28. The related discussions on results and the details on sample fabrications have also been added on Page 23 and 33 in the revised manuscript.

4. In this photoelectrode system, how can the increase of CH₄ flow rate influence the diffusion rate of CH₄ and/or the kinetics surface catalytic reactions?

We thank the referee for his/her insightful question. In simulation results, with the increase of the input CH₄ flow rate from 25 to 50 and 100 mL/min, the diffusion rate of CH₄ increased within the fibrous interwoven pores on mat. We have included the simulation results in the newly added Supplementary Fig. 23 and the related discussions on Page 24 in the revised manuscript.

According to the experimental results, increasing the CH₄ flow rates significantly improved the kinetics of surface catalytic reactions on both activity and selectivity. All contrast

experiments were carried out for 2 hours at a potential of 0.55 V (vs. RHE) and a light intensity of 1 Sun. With the CH₄ flow rate of 50 mL/min, the production rates of CH₃COOH and HCOOH reached 1.86 and 1.20 mmol/g, respectively (7.38 and 6.25 times those at 25 mL/min), with a C₂ product selectivity of 53.9% (referred to as CH₃COOH, 1.26 times that at 25 mL/min). When the CH₄ flow rate was increased to 100 mL/min, the CH₃COOH production rate reached 3.46 mmol/g (13.6 times that at 25 mL/min), with a C₂ product selectivity of 79.7% (1.86 times that at 25 mL/min).

These results further confirmed the significant improvement on the surface catalytic reactions in PEC by enhancing mass transfer kinetics. Based on our free radical activation mechanism mentioned before, the faster flow promoted the migration of methyl radicals for coupling, thereby enhancing the CH₄ conversion as well as the reaction between methyl radicals and HCOOH into CH₃COOH products. Also, the shorter contact time reduced the possibility of unexpected overoxidation reactions to other products such as CO₂, thus improving the C₂ product selectivity. We have included the ¹H NMR spectra of liquid products in these flow reactions with different CH₄ flow rates by R-TiO₂/ZnWO₄ mat in the newly added Supplementary Fig. 29, and supplemented the CO₂ gas production rates in Methods on Page 33 in the revised manuscript. The related discussions and Ref. 46 have been added on Page 24 and 37 in the revised manuscript.

5. What is the reason for the increased selectivity towards acetic acid in the flowing reaction, compared to the batch reaction.

We thank the referee for his/her thoughtful question. Based on our free radical activation mechanism mentioned on Page 22 in the revised manuscript, the faster flow promoted the migration of methyl radicals for coupling, thereby enhancing the CH₄ conversion as well as the reaction between methyl radicals and HCOOH into CH₃COOH products. Also, the shorter contact time reduced the possibility of unexpected overoxidation reactions to other products (*e.g.*, CO₂), thus improving the CH₃COOH product selectivity. This increased selectivity toward CH₃COOH in the flowing reactions by the improved mass-transfer kinetics, were also supported by the simulation and experimental results in our response to Comment 4. The related discussions on the supporting data can be found on Page 24 in the revised manuscript.

REVIEWERS' COMMENTS

Reviewer #1 (Remarks to the Author):

The authors have carefully revised the manuscript according to the comments, and the quality of the manuscript is greatly improved. I recommend the revised manuscript can be accepted for publication.

Reviewer #2 (Remarks to the Author):

The authors have properly addressed all my questions and comments. The manuscript is in good shape. I would suggest the manuscript to be accepted in the journal of Nature Communications.

Reviewer #1 (Remarks to the Author):

The authors have carefully revised the manuscript according to the comments, and the quality of the manuscript is greatly improved. I recommend the revised manuscript can be accepted for publication.

We really appreciate the referee's comments for improved manuscript, as well as the positive evaluation for our work.

Reviewer #2 (Remarks to the Author):

The authors have properly addressed all my questions and comments. The manuscript is in good shape. I would suggest the manuscript to be accepted in the journal of Nature Communications.

We thank the reviewers for their comments on improving the quality of the manuscript, and for their recognition of the revisions and the work.